# Improved Diabetic Wound Healing by EGF Encapsulation in Gelatin-Alginate Coacervates

**DOI:** 10.3390/pharmaceutics12040334

**Published:** 2020-04-08

**Authors:** Seonghee Jeong, ByungWook Kim, Minwoo Park, Eunmi Ban, Soo-Hyeon Lee, Aeri Kim

**Affiliations:** 1College of Pharmacy, CHA University, Seongnam-si 463-400, Gyeonggi-do, Korea; dooblue@hanmail.net (S.J.); eliah1189@naver.com (B.K.); genomw29@naver.com (M.P.); emban@cha.ac.kr (E.B.); 2Immunotherapy Development Team, R&D Division, CHA Biolab, Seongnam 463-400, Korea; esoohaha@chamc.co.kr

**Keywords:** epidermal growth factor (EGF), coacervate, alginate, gelatin, diabetic foot ulcer, drug delivery system, epidermal migration, proinflammatory cytokines

## Abstract

Topical imageplication of epidermal growth fctor (EGF) has been used to accelerate diabetic foot ulcers but with limited efficacy. In this study, we selected a complex coacervate (EGF-Coa) composed of the low molecular weight gelatin type A and sodium alginate as a novel delivery system for EGF, based on encapsulation efficiency and protection of EGF from protease. EGF-Coa enhanced in vitro migration of keratinocytes and accelerated wound healing in streptozotocin-induced diabetic mice with increased granulation and re-epithelialization. While diabetic wound sites without treatment showed downward growth of hyperproliferative epidermis along the wound edges with poor matrix formation, EGF-Coa treatment recovered horizontal migration of epidermis over the newly deposited dermal matrix. EGF-Coa treatment also resulted in reduced levels of proinflammatory cytokines IL-1, IL-6, and THF-α. Freeze-dried coacervates packaged in aluminum pouches were stable for up to 4 months at 4 and 25 °C in terms of appearance, purity by RP-HPLC, and in vitro release profiles. There were significant physical and chemical changes in relative humidity above 33% or at 37 °C, suggesting the requirement for moisture-proof packaging and cold chain storage for long term stability. We propose low molecular weight gelatin type A and sodium alginate (LWGA-SA) coacervates as a novel EGF delivery system with enhanced efficacy for chronic wounds.

## 1. Introduction

There are a number of reports on biomaterials included in wound dressings which accelerate wound healing. For example, He et al. presented a polymer-based nanofibrous wound dressing with antibacterial and wound healing activity [1]. Hybrid polymeric hydrogels were also proposed for wound dressings [2,3]. However, they did not discuss the therapeutic potential of these biomaterials on chronic wound healing. In contrast to normal wound healing, non-healing or chronic wounds such as diabetic foot ulcers show prolonged inflammatory reactions, and correspondingly increased protease response leading to disrupted wound healing by degrading matrix proteins and growth factors [4,5,6,7,8,9,10,11]. Furthermore, the elevated levels of proteases in the wound sites compromise the efficacy of topically applied growth factors in the treatment of chronic wounds [12,13,14,15].

In our previous studies, we aimed to develop a drug delivery system (DDS) for topical delivery of epidermal growth factor (EGF) with improved efficacy in chronic wounds, by encapsulating EGF in complex coacervates. Coacervates composed of high molecular weight gelatin A and sodium alginate (HWGA-SA) showed high encapsulation efficiency for bovine serum albumin (BSA), protecting it from trypsin digestion. However, the encapsulation of EGF was not successful in the same coacervation condition [16]. Formulation screening for coacervates using GA and sodium alginate (SA) was done for EGF encapsulation, by varying the molecular weight of GA, the ratio of GA to SA, and the reaction pH [17]. As previously reported by others, the molecular weight of the polymers was one of the crucial variables [18,19,20]. Precipitation of HWGA-SA at low pH was attributed to strong H-bonding interactions between HWGA and SA. In contrast, low molecular weight gelatin (LWGA) formed homogenous colloidal coacervates with SA with successful EGF encapsulation, suggesting the contribution of electrostatic interactions between EGF and SA.

In the present study, LWGA-SA coacervates encapsulating EGF were evaluated for in vitro activity in keratinocytes, in vivo wound healing efficacy in streptozotocin-induced diabetic mice, and storage stability, to evaluate their potential as a novel therapeutic modality for diabetic foot ulcers.

## 2. Materials and Methods

### 2.1. Materials

Gel strength 90–110 gelatin A (GA, average 20–25 kDa) and trifluoroacetic acid (TFA) were purchased from Sigma–Aldrich (St Louis, MO, USA). Sodium alginate (SA), acetic acid, and acetonitrile (ACN) were purchased from Dae Jung (Gyeonggi-do, South Korea). Epidermal growth factor (EGF) was provided by Daewoong Pharmaceutical Co., Ltd. Dulbecco’s phosphate buffered saline (DPBS) was purchased from Welgene (Gyeongsangbuk-do, South Korea), Dulbecco’s Modified Eagle Medium (DMEM) was obtained from Hyclone (Logan, UT, USA), 0.25% Trypsin-EDTA and fetal bovine serum (FBS) were purchased from Gibco (Paisley, UK).

HaCaT cells were purchased from Lonza (Walkersville, MD, USA). The cell counting kit-8 (CCK-8) was obtained from Dojindo (Kumamoto, Japan). Human EGF antibody as primary antibody (R&D systems, Minneapolis, MN, USA), rabbit anti-mouse horseradish peroxidase (HRP) as secondary antibody (Abcam, Cambridge, UK) were purchased.

### 2.2. Fabrication and Characterization of LWGA-SA Coacervates Encapsulating EGF (EGF-Coa)

Among the two compositions of coacervates previously studied, LWGA-SA (1:0.4) was selected for the present study, based on its higher encapsulation efficiency and better protection from trypsin digestion compared to LWGA-SA (1:1) [17]. Briefly, EGF-Coa was prepared by mixing the aqueous solutions of LWGA (1%, *w*/*w*) and SA (0.5%, *w*/*w*) to (1:0.4), at a total polymer concentration of 5 mg/g, followed by addition of an aliquot of EGF solution, and then finally an aliquot of acetic acid as a pH modifier. LWGA-SA physical mixture (1:0.4) containing the same amount of EGF (EGF-PM) was prepared with the same procedure without the acid titration step. Characterization of the samples is described in the previous paper [17]. Samples were freeze-dried for the in vitro release, in vitro activity, and in vivo tests. To compare the microscopic images of the colloidal coacervates before freeze drying and after rehydration of freeze-dried coacervates, the freeze-dried coacervates composed of LWGA-SA (1:0.4) was rehydrated by adding 100 μL of distilled water. The rehydrated sample was gently rocked at room temperature for one h and its microscopic images were taken with a phase contrast microscope (Leica, Wetzlar, Germany).

For the in vitro release test, Transwell microplates were used as described previously [21]. Release media (700 μL DPBS) was put in each well of a 24-well Transwell microplate (Corning, NY, USA). Freeze-dried samples were loaded in the Transwell inserts (0.4 μm) and 70 μL distilled water was added to hydrate the samples. A control sample was 70 μg EGF solution in 70 μL in the Transwell insert. Aliquots of the release media were collected from each well after incubating the Transwell microplates for given time points (1, 2, 4, 6, 8 h) at 32 °C, 150 rpm. The collected release samples were placed in Eppendorf tubes and stored at 4 °C until analysis. The samples were filtered through a 0.45 μm syringe filter and EGF concentration was analyzed by reverse phase HPLC (Knauer, Berlin, Germany) with a C4 column (00G-4167-E0 Jupiter 5 μm C4 300Å 250 × 4.6 mm, Phenomenex, Torrance, CA, USA) at a column temperature of 40 °C. Mobile phase compositions and the gradient condition are shown in Table 1.

### 2.3. In Vitro Cell Activity Assay

#### 2.3.1. HaCaT Viability Assay

HaCaT keratinocytes were cultured in a 100π dish until 80% confluence in DMEM supplemented with 10% heat-inactivated fetal bovine serum (FBS), penicillin (100 U/mL), and streptomycin (100 μg/mL), in a 5% CO_2_ incubator at 37 °C [22]. The cells were then detached using 0.25% trypsin-EDTA and seeded at 1 × 10^4^ cells per well in a 24-well plate. After incubation at 37 °C for 24 h, culture media were replaced with DMEM containing 0.5% FBS: Media for the negative and EGF groups were replaced with 0.5% FBS 1 mL, 10 ng/mL EGF in 0.5% FBS 1 mL, respectively. Freeze-dried samples were put in the Transwell inserts and 100 μL medium containing 0.5% FBS was added to them. After the inserts were placed in the wells containing 0.5% FBS in DMEM 900 μL, the cells were incubated at 37 °C for 72 h. The proliferation was determined by a CCK-8 assay with absorbance measurement at 450 nm. The results were normalized to the negative group.

The experimental procedures for the hyperglycemic conditions were the same as above except that the wells were replaced with various media containing 25 mM glucose: 1 mL of 0.5% FBS DMEM supplemented with 25 mM glucose for the negative control, 1 mL of free EGF (10 ng/mL EGF) group in 0.5% FBS DMEM, and 900 μL of 0.5% FBS DMEM supplemented with 25 mM glucose for the freeze-dried samples. The glucose concentration of 25 mM for the hyperglycemia control groups was reported previously [23,24,25].

#### 2.3.2. HaCaT Scratch Wound Assay

The experiment was done according to the method published by Li et al. [26]. The mid-line on the back of each well of the 24-well plates was marked with a marker pen. HaCaT cells were cultured in a 100π dish until 80% confluence. After the cells were detached using 0.25% trypsin-EDTA, cells were seeded at a density of 1 × 10^5^ cells per well and incubated at 37 °C overnight. Cells were treated with mitomycin C (2 μg/mL) to inhibit their proliferation for 2 h at 37 °C and 5% CO_2_ [25] of the well with a 200 μL pipette tip, and the wells were washed twice with DPBS to remove debris. 1 mL medium containing 0.5% FBS was added to each well, and pictures were taken with a light microscope (Leica, Wetzlar, Germany). Freeze-dried samples were put in the Transwell inserts, and 100 μL medium containing 0.5% FBS in DMEM was added to them. The inserts were then placed in the wells containing 0.5% FBS in DMEM 900 μL. Culture media contained 0.5% FBS except for the following groups: 0.5% FBS in DMEM 1 mL for the negative, and 10 ng/mL EGF in 0.5% FBS in DMEM 1 mL for the EGF solution samples. After incubation at 37 °C for 24 h, microscopic images were analyzed using Image J (version 1.8, National Institutes of Health, Bethesda, MD, USA) to measure the migration of the cells filling the scratched area. The results were normalized to the negative group as follows:Fold of wound area = {(*A*_0_ − *A*_24_)/*A*_0_}_sample_/{(*A*_0_ − *A*_24_)/*A*_0_}_negative_,(1)
where *A*_0_ is the original wound area and *A*_24_ is the wound area after 24 h.

The experimental procedures for the hyperglycemic conditions were the same as above, except that the wells were replaced with various media containing 25 mM glucose.

### 2.4. In Vivo Study

#### 2.4.1. Streptozotocin (STZ)-Induced Diabetic Mouse Model

Six-week-old male C57BL/6 mice were used for the experiment. All animal procedures were conducted per the Institutional Animal Care and Use Committee of CHA University (IACUC No: 170028). Mice were allowed to acclimate to the environment of a well-ventilated room for a week prior to the experiment. They were fasted for 12–14 h before intraperitoneal injection of STZ (Tocris, Bristol, UK) in 100 mM citrate buffer (pH 4.5). Blood glucose levels after STZ treatment were measured every three to four days with the Accu-Chek active kit (Roche, Basel, Switzerland). Mice with a blood glucose level higher than 250 mg/dL are generally used for the diabetic mouse model [27]. In the present study, a significant rise in the blood glucose level was observed two weeks after two daily STZ treatments at 100 mg/kg. Mice with blood glucose level higher than 250 mg/dL were selected and used for wound healing assay.

#### 2.4.2. Wound Healing Assay in STZ-Induced Diabetic Mouse Model

The diabetic mice were divided into five experimental groups (*n* = 5/group). Under brief anesthesia with isoflurane (Hana pharm, Gyeonggi-do, Korea), the dorsal skin of the mice was shaved and wiped with 70% ethanol. A full-thickness wound (5 mm in diameter) was made on the cleaned dorsal skin of each mouse using a biopsy punch (Kai medical, Oyana, Japan). Group 1 (normal mice) and group 2 (negative control) were given 10 μL distilled water as a control. Group 3 was given 1 μg/10 μL EGF solution. Freeze-dried EGF-PM composed of LWGA-SA (1:0.4) was applied onto group 4. Freeze-dried EGF-Coa composed of LWGA-SA (1:0.4) was applied onto group 5. After applying each freeze-dried sample onto the wounds of groups 4 and 5, DW (5 μL) was dripped to hydrate them. The mice were dosed twice: immediately and three days after wounding. Each mouse was caged separately with food and water. In order to calculate the wound closure rate based on the wound areas, photographs of wounds were taken at different time points after wounding (0, 3, 5, 7 days). The wound healing rate was calculated as follows:Relative wound area (%) = *A_t_/A*_0_ × 100,
where *A*_0_ and *A_t_* are the wound area at time 0 and *t* days after wound, respectively.

Mice were sacrificed using CO_2_ at seven days post-wounding to retrieve the excisional wound samples for histological analysis. Separate groups of mice (*n* = 5/group) were kept for 14 days to monitor the complete wound healing phase. Diabetes induction, mice grouping, wounding procedure, and treatments were the same as those described above except that intraperitoneal injection of ketamine/rompun (3:1) cocktail was used for anesthesia. Photographs of the wound area were taken at different time points (0, 3, 5, 7, 10, 12, 14 days).

#### 2.4.3. Histological Analysis

Tissue samples were fixed with 4% paraformaldehyde (PFA), dehydrated in ethanol, and embedded in paraffin. Paraffin blocks were cut into 5 μm sections for histological analysis. The sections were deparaffinized in xylene and rehydrated through graded ethanol. Hematoxylin and eosin (H&E) and Masson’s trichrome (MT) staining were performed according to the standard protocols. For pan-cytokeratin (PCK) immunostaining, slides were heated in 0.01 M citrate buffer (pH 6.0) for 10 min in a microwave oven, followed by rinsing with phosphate buffered saline (PBS). To eliminate endogenous peroxidases, slides were incubated in 3% hydrogen peroxide, and then incubated in 1% BSA solution for 1 hour. Slides were incubated with primary antibodies overnight at 4 °C and staining accomplished using a horseradish peroxidase-diaminobenzidine (HRP-DAB) staining kit following the manufacturer’s instructions (DAKO). Primary antibodies were anti-pan cytokeratin (diluted 1:20, Abcam, Cambridge, UK). Hematoxylin was used as a counterstain. Images were acquired using a Nikon microscope and analyzed with i-solution software (version 26.1).

To measure the levels of inflammatory cytokines in the wound tissues, a separate group of mice (*n* = 3) were sacrificed at days 3 and 7 after wounding. The excisional wound samples were frozen and cut into small pieces in lysis buffer on ice to prevent thawing, and total RNA was isolated using a cell/tissue miRNA purification kit (Genolution, Seoul, Korea) according to the manufacturer’s instructions. The extracted RNA samples were stored at −80 °C. The total RNA concentration and quality were assessed using a Nano Drop Lite Spectrophotometer 120 V (Thermo Fischer Scientific, Waltham, MA, USA) at the absorbance of 260 and 280 nm. To analyze the miRNA’s expression, cDNA was synthesized using the SuperScript^®^ II Reverse Transcriptase kit (Invitrogen, Carlsbad, CA, USA). For mRNA detection, cDNAs were synthesized. The mRNA levels were analyzed by quantitative real-time PCR (qRT-PCR) using the primers listed in Table 2. The qRT-PCR was assessed as previously reported [28] and was performed on a ViiA™ 7 real-time PCR system (Life Technologies Corporation, Carlsbad, CA, USA) using a Luna universal qPCR master mix (New England Biolabs, Beverly, MA, USA). The 2 ΔΔCq method was used to calculate the expression levels of the mRNA relative to the endogenous control genes 18S ribosomal RNA (18s rRNA).

### 2.5. Stability Test of Freeze-Dried EGF-Coa

To examine the hygroscopicity, freeze-dried samples were placed in desiccators containing saturated salt solutions (LiCl_2_, MgCl_2_, Mg(NO_3_)_2_·6H_2_O, NaNO_2_, NaCl, and KNO_3_ for 11%, 33%, 52%, 64%, 75%, and 93% relative humidity, respectively) [29]. Their appearances were examined after two weeks of storage, and the EGF content was measured by RP-HPLC as described below. To investigate the thermal stability, freeze-dried coacervates were packaged in heat-sealed aluminum pouches to protect from air, light, and moisture. They were then stored in −18, 4, 25, and 37 °C chambers. Samples were visually inspected, and their photos were taken after 4 months of storage. RP-HPLC analysis was done to check the purity of the EGF and in vitro release profiles after storage. In vitro release test was done using Transwell 24-well plates as described above in Section 2.2. For RP-HLC analysis, an AZURA HPLC system (Knauer, Berlin, Germany) with a C18 column (Capcell pack C18, 4.6 μm × 150 mm, 3 μm, Shiseido, Osaka, Japan) was used with the following conditions. Mobile phase was DW in 0.1% trifluoroacetic acid (A) and acetonitrile in 0.1% trifluoroacetic acid (B) with a gradient (A:B = 80:20 at 0, 60:40 at 10, 80:20 at 15 min.). The flow rate was 1.5 mL/min with an injection volume of 20 μL and column temperature at 20 °C. The detection wavelength was UV 280 nm.

## 3. Results

### 3.1. Characterization of EGF-PM and EGF-Coa

Encapsulation of EGF in LWGA-SA coacervates determined by HPLC was 81% and it was protected from trypsin digestion for 2 h [17]. The in vitro release rate of EGF-Coa was significantly slower than that of EGF solution up to 4 h (Figure 1a). EGF-Coa and EGF-PM were different only up to the 1 h time point by two-sample *t*-test. The microscopic images of EGF-Coa before freeze drying and those rehydrated after freeze drying appeared the same (Figure 1b).

### 3.2. In Vitro Activity of EGF-Coacervates

Both EGF-PM and EGF-Coa enhanced the viability of HaCaTs significantly compared to the negative control under normal incubation conditions (Figure 2a). Figure 2b shows the effects of various samples under hyperglycemia conditions. To mimic the hyperglycemia conditions, the HaCaTs were initially incubated for 24 h in media supplemented with 25 mM glucose prior to the cell viability assay by the CCK-8 method. While the activity of EGF solution was lost in hyperglycemia conditions, both EGF-PM and EGF-Coa still showed better viability than the negative control although the significance was less than that in the normal conditions. The representative images of HaCaT scratch wound assay in the normal and hyperglycemic conditions are shown in Figure 2c,d. There were statistically enhanced migration activities of HaCaTs by EGF-Coa treatment compared to the negative group or EGF-solution treatment under normal and hyperglycemic conditions (Figure 2e,f). The EGF solution enhanced the migration significantly compared to the negative control in normal conditions but not in hyperglycemic conditions.

### 3.3. Wound Healing Efficacy of EGF-Coa in Diabetic Wound Model in Mice

The efficacy of EGF solution, EGF-PM, and EGF-Coa on wound closure was compared in the full-thickness excisional wounds using the STZ-induced diabetic mouse model. When the freeze-dried EGF-PM and EGF-Coa were applied to the wound site and wetted with 5 μL DW, they immediately formed hydrated film (Figure 3a). Wound closing was effectively accelerated after treatment with EGF-Coa throughout the test period compared to the negative control, free-EGF, or EGF-PM groups (Figure 3b). Statistical analysis shows significant improvement in wound healing by EGF-Coa compared to the free-EGF or EGF-PM groups (Figure 3c). The wound size of the negative control group on day 3 became larger than the other groups, including those of the free-EGF or EGF-PM groups. The wound healing rate of the negative group was significantly slower than that of the normal (non-diabetic) mice. The wound healing rate of the EGF-Coa group was not significantly different from that of the normal control group and was faster than those of the negative, free-EGF, and EGF-PM groups. Complete wound closure was seen in all groups by day 14.

### 3.4. Histological Analysis

Seven days after wounding by biopsy punch, hematoxylin and eosin (H&E), Masson’s trichrome (MT), and pan-cytokeratin (PCK) staining were performed to evaluate the effect of EGF-Coa on wound healing parameters, including granulation and re-epithelialization [30]. Epithelial cell differentiation activity in vivo was analyzed by PCK, which visualizes the epithelial tissue stained in brown. Figure 3 shows the histological images of representative samples from each group. Black arrows in MT images and red arrows in PCK are to indicate the front of horizontally migrating epidermis over freshly formed matrix. Normal mice (Figure 4a) showed new tissue formation in the wound area and epidermis migrating along the surface of freshly deposited dermal tissue. There was poor matrix formation in the negative group (diabetic mice without treatment) (Figure 4b). Downward growths of hyperproliferative epidermis along the wound edges without horizontal migration can be seen in the negative group. Treatment with an EGF solution or EGF in physical mixture of LWGA and SA was helpful to rebuild the dermal matrix and regain horizontal migration of epidermis but not as much as the normal group (Figure 4c,d). EGF-Coa treatment resulted in horizontal migration of epidermis over the newly deposited dermal matrix, similar to the normal group. (Figure 4e). MT and PCK images of all samples are presented in the Appendix A, respectively. Table 3 summarizes the number of the horizonal migration fronts of epidermis above the freshly formed matrix in the MT and PCK images shown with the arrows in Appendix A. Among ten wound edges examined, there were eight horizontal migrations in the normal- and EGF-Coa-treated diabetic wound, which was two folds of those in the negative group. Numbers in EGF in solution or physical mixtures of LWGA and SA were in between those of the normal and the negative groups.

### 3.5. Inflammatory Cytokine Levels

There was no death or significant changes in the body weight during the experiment except two mice which had lower body weight than the others at dosing and died at 4 and 7 days after dosing, respectively. The body weight data are shown in the Appendix A. There were no significant differences among the groups in the levels of proinflammatory cytokines IL-1b, IL-6, and TNF-α on day three (Figure 5). The cytokines levels in the EGF-PM and EGF-Coa groups on day seven were lower than those on day three, whereas the negative group did not show significant changes between day three and day seven. The EGF-Coa group appeared to show the lowest level of all three cytokines on day seven, although a statistical analysis was not feasible due to the limited number of samples.

### 3.6. Stability of Freeze-Dried EGF-Coa

Freeze-dried samples stored above 33% relative humidity (RH) showed physical (Figure 6a) and chemical changes after two weeks of storage at room temperature (Table 4). There were no changes in the appearance of the freeze-dried EGF-Coa after four months of storage at −18, 4, and 25 °C except at 37 °C (Figure 6b). The RP-HPLC showed massive degradation peaks of EGF-Coa from 37 °C storage, whereas samples stored at lower temperatures appeared intact (Figure 6c). In vitro release patterns of various storage temperatures were consistent with the findings from the HPLC analysis (Figure 6d).

## 4. Discussion

Previously, we explored various compositions of LWGA-SA to effectively encapsulate EGF at lower pH [17]. Homogenous coacervation with polydispersity index (PDI) below 0.4 and turbidity above 1.5 were selected to prepare EGF-Coa with improved encapsulation efficiency. PDI values less than 0.2 are typically considered monodisperse, while those higher than 0.7 are considered highly polydisperse [18]. In the present study, the in vitro and in vivo assays were done with EGF-Coa composed of LWGA-SA (1:0.4) to evaluate their potential as a novel therapeutic modality for chronic wound healing. This particular composition was chosen because of its higher encapsulation efficiency (81%) and better protection against trypsin digestion than other compositions tested [17]. The freeze-dried coacervates instead of colloidal coacervates were used for the in vitro and in vivo studies to ensure their long-term storage. When the freeze-dried coacervates were hydrated, they returned to the colloidal coacervates. Although the in vitro release rate of EGF-Coa was slower than that of EGF solution up to the 4 h time point, the difference between EGF-Coa and EGF-PM was significant only up to the 1 h time point. EGF-Coa protected EGF from trypsin digestion more effectively than EGF-PM up to 2 h time point but not at 4 h time point (data not shown). These data indicate that the EGF encapsulation in the coacervate is effective up to 2 h in vitro.

Successful wound healing requires complex interactions and cross-talk between fibroblasts, keratinocytes, endothelial cells, immune cells, and others [10]. Fibroblasts play an important role in wound healing by enhancing the re-epithelialization and remodeling of the extracellular matrix [31,32]. EGF stimulates fibroblasts to migrate into the wound site and proliferate in order to reconstitute the various connective tissue components [31]. Although there is no single in vitro model to mimic the complexity of chronic wound healing in vivo [32], Rowe et al. and Hehenberger et al. demonstrated the inhibition of proliferation and resistance to EGF in the fibroblasts under hyperglycemic conditions [33,34]. Our previous study showed that EGF-Coa could significantly enhance the migration of human dermal fibroblasts (HDFs) compared to EGF solution (*p* < 0.05), and marginally compared to EGF-PM (*p* < 0.01) [17]. In addition to HDF, keratinocytes are important in the re-epithelialization process and have EGFR, a direct target of EGF [10]. Therefore, in the present study, we evaluated the effects of EGF as a solution, EGF-PM, and EGF-Coa on the viability and migration of keratinocytes of the HaCaT cell. For migration, we used a wound scratch assay of keratinocytes, a widely used in vitro assay to study re-epithelialization [10]. HaCaT cell under hyperglycemia conditions (25 mM glucose) was used to mimic diabetic wounds in vitro [23,24,25]. The HaCaT cells showed better viability in hyperglycemia conditions when treated with EGF-PM or EGF-Coa than the negative control or EGF solution. Both EGF-PM and EGF-Coa induced significantly higher activity in HaCaT cell migration than the negative, EGF solution, or vehicle control groups. There were statistically enhanced migration activities of HaCaTs by EGF-PM and EGF-Coa treatment compared to the negative group or EGF-solution treatment under normal and hyperglycemic conditions (Figure 1e,f).

Immediate film formation upon application of the freeze-dried EGF-Coa to the wound site demonstrates its advantage in terms of accurate and easy dosing (Figure 3a). Complete wound closure of all mice groups (normal control and diabetic mice treated or not treated) by day 14 is consistent with other in vivo mice models reported previously (Figure 3b,c) [30,35,36]. While the wound closure in the negative control group of STZ-induced diabetic mice was slower than that in the normal mice, the EGF-Coa significantly accelerated wound closure in the STZ-induced diabetic mice compared to all the other groups up to 7 days. Enhanced wound healing rate during the earlier phase in mice can be beneficial because it can lead to less chance of complication when translated to clinical application. The in vivo efficacy of wound healing correlates with the in vitro activity in that EGF-Coa was better than the negative or EGF solution groups. However, statistically better in vivo wound healing of EGF-Coa compared to EGF-PM cannot be predicted from the keratinocytes’ viability or migration assay results shown in Figure 2. Notably, the better activity of EGF-Coa compared to the EGF-PM was observed on the migration of HDF in our previous study. These conflicting in vitro and in vivo results are another example of complex mechanisms for chronic wound healing.

Chronic wound healing is characterized by poor matrix formation and the thick and hyperproliferative epidermal edges with mitotically active keratinocytes unable to migrate along the surface, moving down deep into the neodermis [10,11]. In contrast to the normal group with new tissue formation in the wound area and epidermis migrating along the surface of newly deposited dermal tissue, the negative group showed poor matrix formation and downward growth of epidermis instead of horizontal migration synthesis (Figure 6a,b). Such histological images confirm the non-healing nature of the wounds in STZ-treated diabetic mice. Improved efficacy of EGF-Coa in vivo was also demonstrated by histological analysis. There was the increased synthesis of dermal matrix and horizontal migration of epithelial tissues in the diabetic wounds treated with EGF-Coa, similar to the normal mice. These histological results are consistent with the enhanced migration of HDF by EGF-Coa observed in our previous study and the improved migration of HaCaT cells under hyperglycemic conditions by EGF-Coa in the present study. When the epidermal fronts migrating above the newly formed matrix were estimated in the MT and PCK images, the normal- and EGF-Coa- treated diabetic wounds showed a higher number than EGF-solution- or physical-mixture-treated diabetic wounds (Appendix A, and Table 3). These results demonstrate that EGF-Coa was effective in restoring horizontal migration of epidermis to cover the surface of the newly formed matrix in the wound area of the diabetic mice. Better efficacy of EGF-Coa than EGF-PM in terms of horizontal migration of epidermis in vivo correlates well with the wound size measurement. Further study is warranted to better understand the beneficial effects of encapsulation of growth factors at the cellular and molecular levels.

The levels of proinflammatory cytokine levels in the wound sites were similar among all test groups on day 3, but the EGF-Coa treatment group showed lower levels on day 7. Although the proinflammatory response is an important step for normal wound healing, a prolonged inflammation phase leads to the non-healing nature of chronic wounds such as diabetic foot ulcers [6,7,10,11,14,32]. The high level of proinflammatory cytokines in the negative group on day 7 corroborates that the full-thickness excised wounds in the present STZ-treated mice had characteristics of chronic wounds. Siqueira reported an increased level of TNF-α in diabetic wounds in mouse models in association with fibroblast apoptosis [37]. In the same model, the TNF-α inhibition by pegsunercept resulted in improved healing with increased fibroblast density. Kim et al. showed that decreased levels of TNF-α and IL-1 after treatment with EGF and hyaluronate-EGF conjugate on the skin wound in normal (non-diabetic) rats [15]. Gainza et al. reported improved efficacy of EGF-loaded solid lipid nanoparticles (SLN) in a type 2 diabetic mice model [36]. They reported the resolution of inflammatory response after eight days of treatment of various SLN based on semi-quantitative analysis of H&E staining, but they did not present the cytokine levels. These previous reports shed light on the correlations found in the present study between the diabetic wound healing and the proinflammatory cytokine levels in the wound beds on day 7. The negative group with elevated levels of proinflammatory cytokines did not heal well. In contrast, the EGF-Coa group with a lower level of proinflammatory cytokines showed horizontal migration of epidermis over the newly formed matrix similar to the normal group. To the best of our knowledge, the present study is the first report on the indirect but beneficial effect of EGF on reducing the proinflammatory cytokines in diabetic or chronic wounds. Although EGF is not a direct modulator of inflammation, accelerated wound healing by EGF-Coa might have positively contributed to lowering the inflammatory reactions in diabetic wounds. The results warrant further mechanistic study to extrapolate to translational research. Taken together, the EGF-Coa with in vitro HDF and keratinocytes activity higher than EGF solution, demonstrated its beneficial effects in not only the accelerated wound healing rate accompanied by horizontal migration of epidermis but also the reduced level of proinflammatory cytokines.

Recently, Choi et al. proposed structural modifications of EGF and bFGF molecules to enhance their activity for diabetic wound healing after topical application in hyaluronic acid and poloxamer matrix [35]. The modified growth factors showed improved conformational and chemical stability in solution and during storage, but their stability against proteases was not measured. It would be an interesting future research subject to compare the structurally modified growth factors and the EGF-Coa, in terms of proteolytic degradation and their activity in vitro and in vivo. An even more interesting study would be to encapsulate those modified growth factors in coacervates for added or synergistic effects for enhanced wound healing. Berlanga-Acosta et al. pointed out the controversies over the clinical efficacy of topical EGF in treating diabetic foot ulcers. They proposed an intralesional injection of EGF as an alternative [32]. However, an intralesional injection may not be as desirable as a topical application for the patients as well as clinicians. Yet, it would be interesting to compare the efficacy of an intralesional injection and topical application of EGF-Coa. EGF-loaded solid lipid nanoparticles (SLN) proposed by Gainze et al. showed positive results in enhancing EGF activity in the diabetic wound [36]. Although they did not present the data on the protection of EGF from proteases by encapsulation in SLN, one can speculate that SLN might have protected EGF from proteases as EGF-Coa does. One disadvantage of SLN could be the manufacturing process requiring organic solvents and centrifugation steps. EGF-Coa in the present study could be the very first topical delivery system with feasibility for scale-up and clinical application.

## 5. Conclusions

The LWGA-SA-coacervate-based delivery of EGF showed better activity than free-EGF not only in the in vitro keratinocytes scratch wound assay but also in the in vivo diabetic mouse wound model. EGF-Coa was beneficial not only in direct wound healing but also in lowering proinflammatory cytokines. For long-term storage, moisture-proof packaging and cold chain storage will be required. We propose EGF-Coa as a novel EGF delivery system to accelerate the wound healing process in chronic wounds such as diabetic foor ulcer (DFU). The same approach will be useful to enhance the efficacy of fibroblast growth factor (FGF) or platelet derived growth factor (PDGF) for diabetic foot ulcers.

## Figures and Tables

**Figure 1 pharmaceutics-12-00334-f001:**
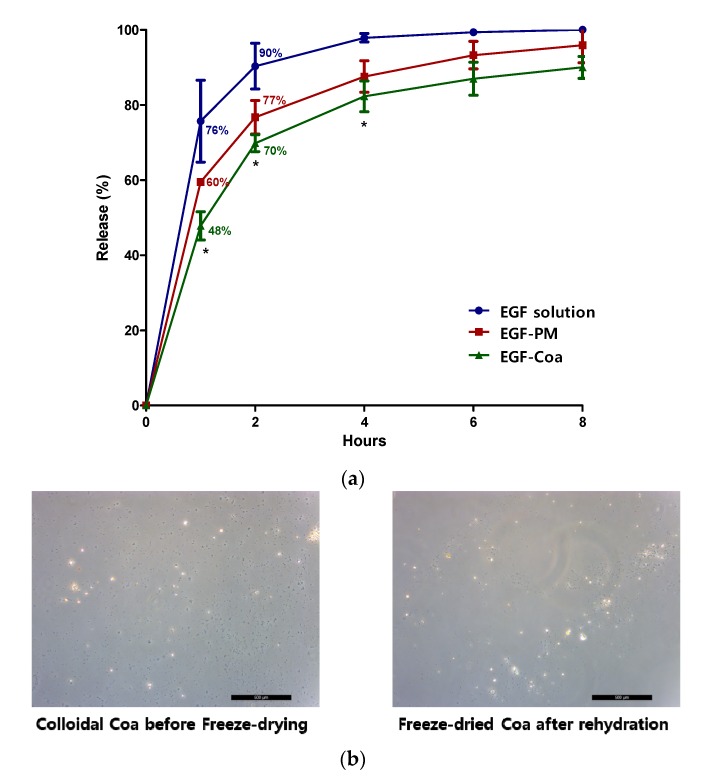
(**a**) In vitro release profiles of epidermal growth factor (EGF) solution, EGF-PM, and EGF-coacervate (EGF-Coa) at 32 °C determined by HPLC. The statistical difference was determined by one-way ANOVA with GraphPad Prism software (version 5.01). (**b**) Comparison of microscopic images of low molecular weight gelatin type A and sodium alginate (LWGA-SA) (1:0.4) coacervates before freeze drying and after rehydration of freeze-dried coacervates after one hour rehydration (Scale bar = 500 μm).

**Figure 2 pharmaceutics-12-00334-f002:**
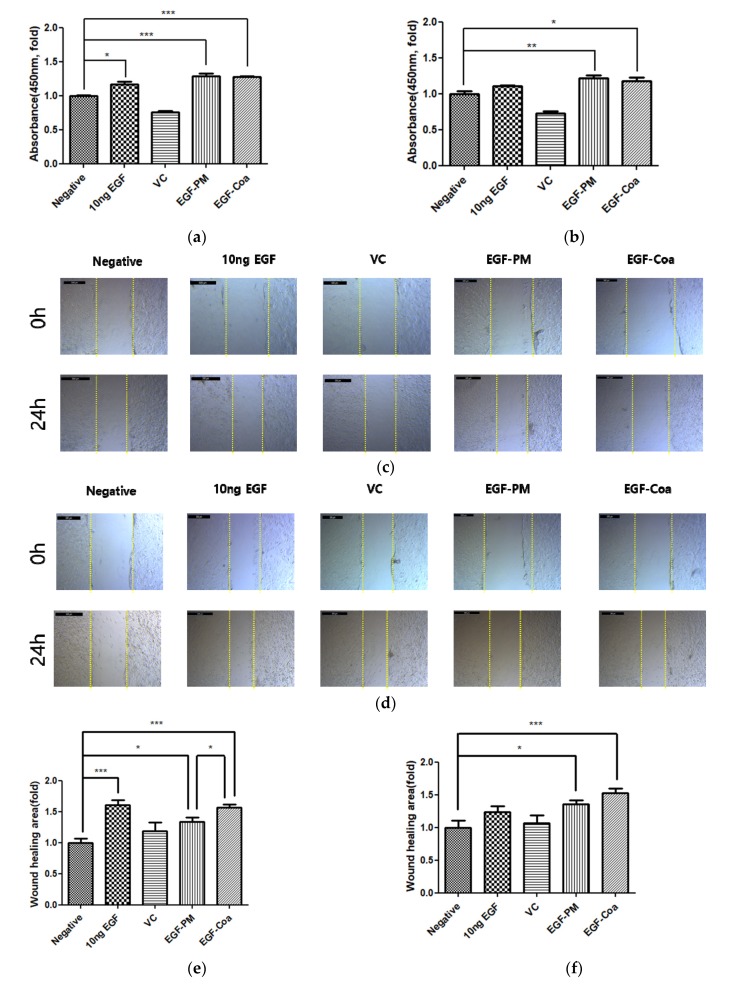
In vitro activity of EGF on HaCaT cells after treatment of various samples. HaCaTs viability in (**a**) normal and (**b**) hyperglycemic conditions. VC stands for the mixture of LWGA-SA (1:0.4) without EGF. Representative images of HaCaTs after scratch wound in (**c**) normal and (**d**) hyperglycemic conditions. Statistical analysis of migration (**e**) in normal and (**f**) hyperglycemic conditions. * *p* < 0.05, ** *p* < 0.01, *** *p* < 0.001. Two-way ANOVA was done with GraphPad Prism software.

**Figure 3 pharmaceutics-12-00334-f003:**
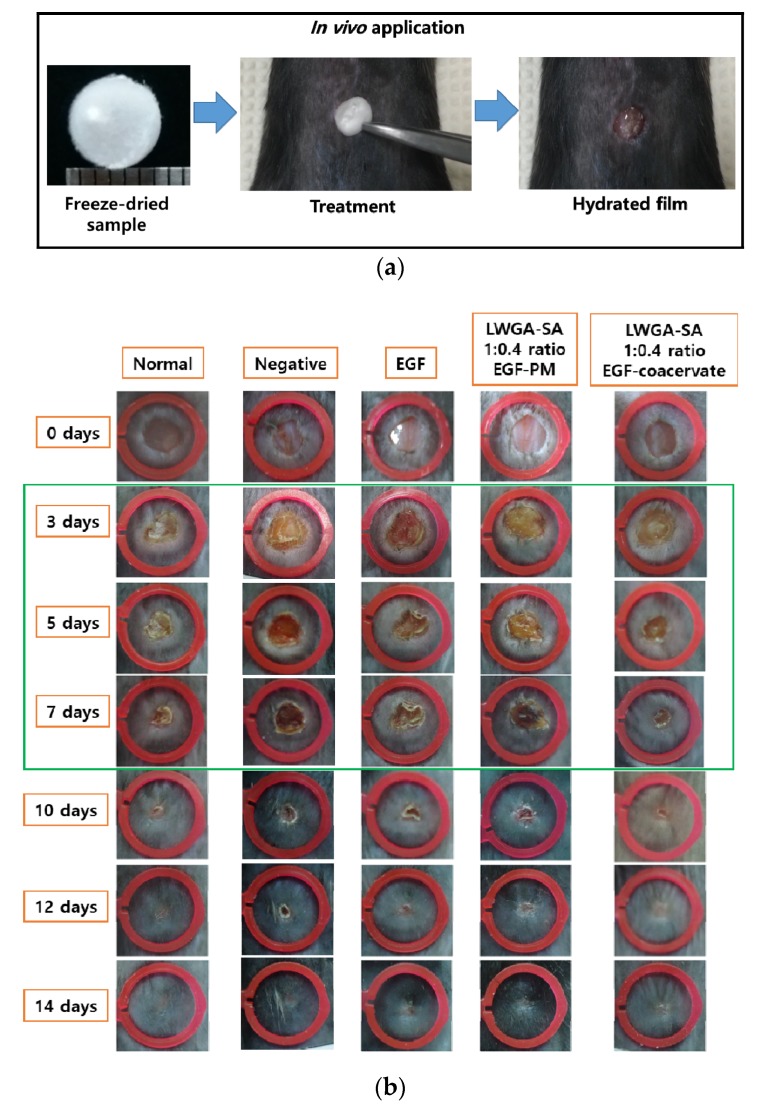
Effect of EGF-Coa on full-thickness excisional wounds in streptozotocin (STZ)-induced diabetic mice model compared to normal mice. (**a**) In vivo application of freeze-dried EGF-coacervate to the wound site. (**b**) Representative photographs of ① Normal: wound sites in normal mice treated with Dulbecco’s phosphate buffered saline (DPBS), ② Negative: wound sites in STZ diabetic mice treated with DPBS, ③ Free-EGF, EGF-PM, and EGF-coacervate: wound sites in STZ diabetic mice treated with EGF solution, EGF-PM, and EGF-coacervate. (**c**) Average and standard deviations of wound area over time, measured as the percentage of the wound area at time 0. The results are shown as combined data from Method 2.3 (0, 3, 5, 7 days, *n* = 10) and (0, 3, 5, 7, 10, 12, 14 days, *n* = 5). * *p* < 0.05, ** *p* < 0.01. Two-way ANOVA was done with GraphPad Prism software.

**Figure 4 pharmaceutics-12-00334-f004:**
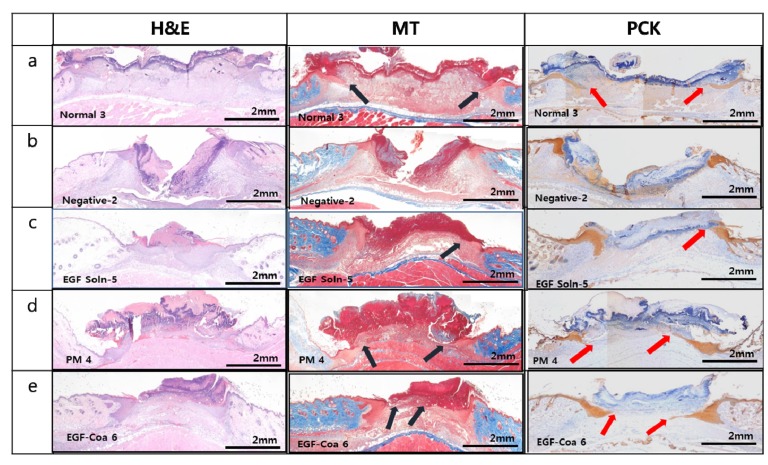
Histological images of wound beds with hematoxylin and eosin (H&E), Masson’s trichrome (MT), and pan-cytokeratin (PCK) staining. (**a**) Normal (non-diabetic mice), (**b**) Negative (diabetic mice) treated with PBS, (**c**), (**d**), and (**e**) Diabetic mice treated with EGF solution, EGF-PM, and EGF-Coa. Black (MT) and red (PCK) arrows indicate the pointed fronts of horizontal migration of keratinocytes along the surface of newly formed granulation tissue.

**Figure 5 pharmaceutics-12-00334-f005:**
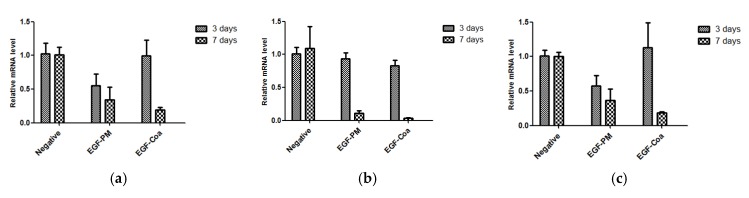
Proinflammatory cytokine levels. (**a**) IL-6, (**b**) IL-1, and (**c**) TNF-α in the wound area of STZ-induced diabetic mice after treatment with EGF-PM and EGF-Coa. 3 days: Negative, *n* = 3; EGF-PM, *n* = 4; EGF-Coa, *n* = 4. 7 days: Negative, *n* = 3; EGF-PM, *n* = 3; EGF-Coa, *n* = 2.

**Figure 6 pharmaceutics-12-00334-f006:**
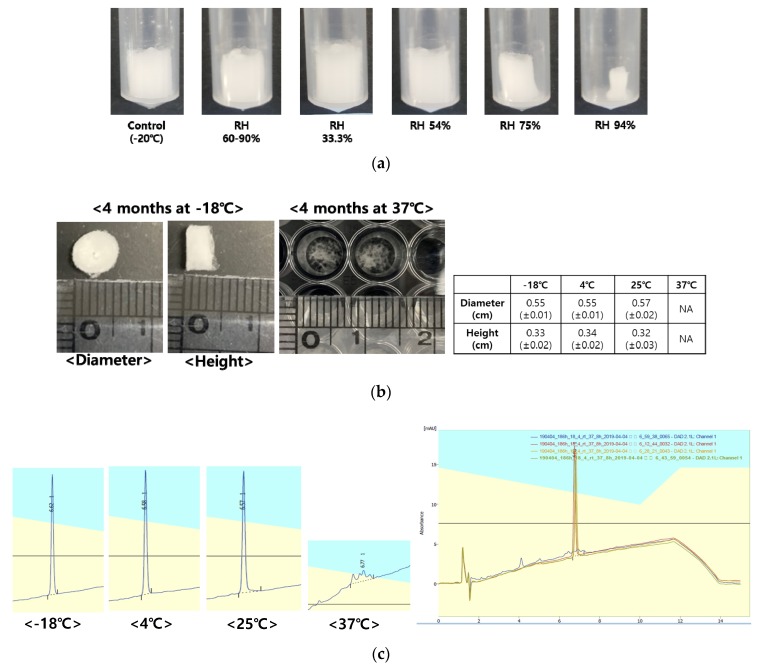
Stability of freeze-dried EGF-Coa. (**a**) Appearances after 2 weeks of storage at various relative humidity at room temperature, (**b**) appearances and dimensions after 4 months of storage at −18, 4, 25, and 37 °C, (**c**) representative HPLC chromatograms of samples after 4 months of storage at −18, 4, 25, and 37 °C, and the overlay plots of the same samples, (**d**) in vitro release profiles of samples after 4 months of storage at −18, 4, 25, and 37 °C.

**Table 1 pharmaceutics-12-00334-t001:** Gradient profile of mobile phase. Solution A: 0.1% trifluoroacetic acid (TFA) in distilled water, Solution B: 0.1% TFA in acetonitrile.

Elution Time (min)	Solution A (%)	Solution B (%)
4	95	5
12	80	20
14	80	20
22	65	35
25	65	35
30	80	20
35	95	5
55	End of the run	

**Table 2 pharmaceutics-12-00334-t002:** List of primers.

Gene	Forward	Reverse	Size (bp)
IL-6	TGGTGACAACCACGGCCTTC	GCCTCCGACTTGTGAAGTGGT	104
IL-1b	GCTGTGGAGAAGCTGTGGCA	GGGAACGTCACACACCAGCA	153
TNF-α	GACAAGGCTGCCCCGACTACG	CTTGGGGCAGGGGCTCTTGAC	110
18s rRNA	GCAATTATTCCCCATGAACG	GGCCTCACTAAACCATCCAA	111

**Table 3 pharmaceutics-12-00334-t003:** Counts of the horizontally migrating fronts representing re-epithelialization.

Sample	MT	PCK
Normal	8/10	8/10
Negative	4/10	4/10
EGF solution	5/10	5/10
PM	6/10	6/10
EGF-Coa	8/10	8/10

Numbers represent the total counts of arrows in MT and PCK over 10 wound areas (raw data in Appendix A).

**Table 4 pharmaceutics-12-00334-t004:** Changes in HPLC area % relative to −20 °C storage sample after 2 weeks of storage in various relative humidity at room temperature.

Sample	Average Area % (*n* = 2)	Range
−20 °C (control)	100.0	98.8, 101.2
11.1% RH	99.6	97.7, 101.5
33.3% RH	92.8	93.3, 92.2
54% RH	93.0	93.4, 92.6
75% RH	88.2	89.2, 87.1
94% RH	67.4	69.4, 65.3

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
