# Peer review of "Improved Diabetic Wound Healing by EGF Encapsulation in Gelatin-Alginate Coacervates"

_pharmaceutics, 2020, doi:10.3390/pharmaceutics12040334_

Round 1

Reviewer 1 Report

This manuscript proposed a gelatin-alginate coacervate for encapsulation and protection of epidermal growth factor (EGF), and this complex coacervate (EGF-Coa) demonstrated enhanced performance in vitro migration of keratinocytes and accelerated the diabetic wound healing process. Indeed, this method provides an efficient strategy for EGF delivery system toward chronic wounds under effective storage condition. However, before it can be published in this journal, there are some issues should be addressed as below:

  1. Please add the basic fabrication procedure of EGF-Coa, and information about EGF-PM.
  2. Basic characterization should be applied for the colloidal coacervates.
  3. How about the encapsulation capacity and EGF release profile of this complex coacervates?
  4. In the experiment of stability of EGF-Coa, all the samples were pre-freeze-dried before placed in desiccators. I wondered if the stability result of these freeze-dried sample could be consistent with the result of colloidal coacervates.
  5. All the figures should be revised into a uniform format.
  6. Scale bar is missing in Figure 3 and figures in Support information.
  7. Some other related work about biomaterials as wound dressing might be cited and discussed to improve their manuscript. For example: Chemical Engineering Journal, 2020, 385, 123464; Biomaterials, 2010, 31, 1, 83-90; Carbohydrate Polymers, 2019, 226, 115302.

Reviewer 2 Report

This manuscript describes the research on topical application a complex coacervate (EGF-Coa)  composed of the low molecular weight gelatin type A and sodium alginate as a novel delivery system for EGF. These studies are a continuation of the experiments described in the article published in 2019 in Pharmeceutics. This is a quite interesting field, however, there is a number of issues that should be clarified is several aspects.

1.For clarification I want to notice that Cell Counting Kit-8 (CCK-8) allows for the determination of general cell viability but not proliferation. It is based on activity of the metabolic activity (dehydrogenases)  and therefore is a marker reflecting viable cell metabolism and not specifically cell proliferation.

2. Authors show that EGF-PM and EGF-Coa enhanced proliferation of HaCaTs significantly compared to the negative control under the normal and hyperglycaemic incubation condition. If so such stimulation may probably affect the results of migration. To differentiate the  contributions  of  cell  proliferation  and  migration  to  wound  closure,  cell  cycle  blocker  hydroxyurea can be added in the scratch wound model as it was for example demonstrated with HaCaT cells in the paper by Weglowska et al. I would like authors to repeat migration experiments under such conditions
(Proangiogenic properties of nucleoside 5'-O-phosphorothioate analogues under hyperglycaemic conditions. Curr Top Med Chem. 2015;15(23):2464-74).

3. I would like to see deeper discussion between results obtained by authors with HDFs and HaCaTs

4. What was the mM concentration of glucose in control Dulbecco Modified Eagle Medium (DMEM) used by authors?

Round 2

Reviewer 1 Report

I am generally satisfied with the revisions, and I recommend the acceptance of this work.

Reviewer 2 Report

Authors improved the manuscript as requested.